# Stenting versus balloon angioplasty alone in patients with below-the-knee disease: A propensity score-matched analysis

**Jihun Ahn**[1☯], **JinSu Byeon**[2☯], **Byoung Geol Choi**[3‡], **Se Yeon Choi**[4], **Jae Kyeong Byun**[4], **Jinah Cha**[4], **HyeYon Yu**[5], **Cheolho Lee**[1], **Jin Oh Na**[6], **Cheol Ung Choi**[6], **Eung Ju Kim**[6], **Chang Gyu Park**[6], **Hong Seog Seo**[6], **Woong-gil Choi**[1], **Seung-Woon Rha**[6‡*]

1 Cardiovascular Center, Konkuk Univesrity Chungju Hospital, Chungju, Korea, 2 Department of Internal Medicine, Soonchunhyang University Gumi Hospital, Gumi, Korea, 3 Cardiovascular Research Institute, Korea University, Seoul, Korea, 4 Department of Medicine, Korea University Graduate School, Seoul, Korea, 5 School of Nursing, College of Medicine, Soonchunhyang University, Asan, Korea, 6 Cardiovascular Center, Korea University Guro Hospital, Seoul, Korea

☯ These authors contributed equally to this work.
‡ SWR and BGC also contributed equally to this work.
* swrha617@yahoo.co.kr

**Data Availability Statement:** All relevant data are within the paper and its Supporting Information files.

## Abstract

Percutaneous transluminal angioplasty (PTA) is considered an effective treatment in patients with critical limb ischemia (CLI). However, the long-term durability of below-the-knee (BTK) PTA is known to be limited. This study sought to compare the 1-year clinical outcomes following stenting versus balloon angioplasty alone in BTK lesions. This study included 357 consecutive patients (400 limbs, 697 lesions) with BTK lesions who underwent PTA from September 2010 to December 2016. All enrolled patients were treated either by stenting (stent group; 111 limbs of 102 patients) or plain old balloon angioplasty (POBA group; 289 limbs of 255 patients). Stent group includes both primary and provisional stenting. Angiographic outcomes, procedural success, complications, and clinical outcomes were compared between the two groups up to 1 year. After propensity score matching (PSM) analysis, 56 pairs were generated, and the baseline and angiographic characteristics were balanced. The procedural success and complications were similar between the two groups; however, the incidence of procedure-related perforation was higher in the POBA group than in the stenting group [5(11.9%) vs.1 (0.9%), P = 0.009]. Six- to 9-month computed tomography or angiographic follow-up showed similar incidences of binary restenosis, primary patency, and secondary patency. In the 1-year clinical follow-up, there were similar incidences of individual hard endpoints, including mortality, myocardial infarction, limb salvage, and amputation rate, with the exception of target extremity revascularization (TER), which tended to be higher in the stenting group than in the POBA group [21 (20.8%) vs. 11 (10.9%), P = 0.054]. Although there was a trend toward a higher incidence of TER risk in the stenting group, stent implantation, particularly in bail-out stenting seemed to have acceptable 1-year safety and efficacy compared to POBA alone in patients undergoing BTK PTA.

**Funding:** This work was supported by the Korea Medical Device Development Fund grant funded by the Korea government (the Ministry of Science and ICT, the Ministry of Trade, Industry and Energy, the Ministry of Health & Welfare, the Ministry of Food and Drug Safety) (Project Number: 9991006707, KMDF_PR_20200901_0034).

## Introduction

Since the report of the Transatlantic Inter-Society Consensus (TASC) II guideline in 2007, endovascular therapy has become one of the first-line treatment strategies for patients with critical limb ischemia (CLI) [1]. Moreover, some multicenter randomized controlled trials have shown that the amputation-free survival rate following percutaneous transluminal angioplasty (PTA) is similar to that of bypass surgery [2, 3]. However, a similar limb salvage rate of PTA was achieved in the treatment of CLI with below-the-knee (BTK) lesions when compared with surgery. Nevertheless, the higher rate of target lesion revascularization (TLR) and the lower primary patency rate are still causing concerns [4].

With the accumulation of experience and evidence, interventional treatment has largely replaced the surgical treatment worldwide in the treatment of CLI with BTK lesions [5, 6]. Moreover, with the development of device technology, various devices and methods have been newly adopted in the treatment of CLI patients with BTK lesions, such as conventional balloons, drug-coated balloons (DCBs), bare metal stents (BMSs), and drug-eluting stents (DESs) [7–10]. Despite improvements in device technology and interventional techniques, limited rates of primary patency and TLR remain unsolved problems.

Previous research related to BTK intervention comparing plain old balloon angioplasty (POBA) with stent implantation has been plagued by relatively small sample sizes, which might have affected the study results [11–14]. In addition, despite the accumulation of well-designed randomized comparative studies on stent implantation and POBA in the treatment of CLI, high level of evidence is still limited [15]. Moreover, although several studies and strategies for BTK intervention have been evaluated, including DCB and stent implantation, there is no consensus on the effectiveness of such treatment strategies yet. We believe the armamentarium of BMS compared to conventional balloons, particularly in bail-out situations. And, we also predict that we can prove the safety and efficacy of BMS in the treatment of BTK lesions in a relatively large number of patients. This study aimed to retrospectively compare the safety and efficacy of stenting (bail-out or planned as primary stenting) with standard POBA in BTK lesions for up to 1 year.

## Materials and methods

### Study design

A total of 357 consecutive CLI patients with BTK lesions who underwent PTA between September 2010 and December 2016 at the Cardiovascular Center of Korea University Guro Hospital, Seoul, South Korea, were enrolled in this study. Patients were stratified into two groups according to procedural type: the stent group (patients treated with stent implantation; n = 102) and the POBA group (patients treated with POBA alone; n = 255). All procedures were reviewed and approved by the institutional review board of Korea University Guro Hospital, and all patients or their legal guardians were given a thorough written and verbal explanation of the study procedures before giving written consent for participation in this study.

Inclusion criteria for the study groups were as follows: (1) CLI documented as Rutherford stage 4, 5, or 6, as evidenced by resting pain, ulceration or gangrene; and (2) identification of hemodynamically significant BTK stenosis (>70% by visual estimation) on imaging studies. The exclusion criteria were as follows: acute limb ischemia, history of severe contrast allergy, hypersensitivity to aspirin and/or clopidogrel, previous use of stents, sepsis, and life expectancy of less than 12 months. All enrolled patients were divided based on stent implantation (the stent group vs. the POBA group). The stent group included either elective stent implantation (primary stenting following acceptable balloon response) or bail-out stent implantation due to significant immediate recoil or flow-limiting dissection.

## Procedural techniques and antiplatelet regimen

Loading doses of clopidogrel (300–600 mg) and aspirin (200–300 mg) were administered before the index procedure. After obtaining the arterial access site, a bolus dose of unfractionated heparin (70–100 units/kg) was administered after sheath insertion. Following PTA, all patients received aspirin (100 mg) and clopidogrel (75 mg) as maintenance dual antiplatelet regimens for at least a month. Cilostazol (100 mg) was prescribed twice daily based on the physician's discretion for 1 month, 3 months or 6 months.

PTA procedures were performed following standard techniques. After successful 0.014inch guidewire crossing either intraluminally or subintimally, prolonged balloon dilation was performed for 2 to 3 minutes. Selective cases were safely recanalized by the transcollateral approach or retrograde approach, particularly following unsuccessful reentry to the distal true lumen after subintimal wiring. Balloon dilatation was performed using low-profile balloons with diameters ranging from 2 mm to 3 mm, and lengths ranging from 20 mm to 220 mm, based on the reference vessel size. The stent group was further stratified into two groups based on the strategy (provisional stent implantation vs. elective stent implantation). Provisional stent implantation was performed when a complication occurred, such as flow-limiting dissection and persistent recoil, even after repeated prolonged balloon dilatation. Elective stent implantation was performed despite a favorable balloon response as 'primary stenting' in selected patients when they were enrolled in a multicenter stent study that compared the efficacy and safety between stent implantation and POBA in the treatment of BTK lesions. Non-drug-eluting BMS Maris Deep (Medtronic Inc., Minneapolis) self-expanding nitinol stent, Chromis Deep (Medtronic Inc., Minneapolis) balloon expandable stents, and Xpert (Abbott Vascular, Abbott Park, IL) self-expanding nitinol stents were used in the stent group.

## Study endpoints and definition

Quantitative vascular angiographic parameters were measured and analyzed before PTA, immediately after PTA, and 6–12 months after the index procedure. Quantitative vascular angiographic measurements and analyses were performed at the Cardiovascular Center of Korea University Guro Hospital, Seoul. Computed tomography (CT) occasionally replaced angiography based on the physician's decision. The primary angiographic endpoint was the incidence of binary restenosis at 6–12 months. Secondary angiographic endpoints were primary and secondary patency at 6–12 months after the index procedure. Primary patency was achieved whenever significant flow-limiting restenosis did not develop without any additional repeat intervention on invasive angiography or CT angiography. Secondary patency was defined as patency achieved after reintervention for significant restenosis of the treated artery; it was also assessed on invasive angiography or CT angiography after the reintervention procedure.

One year after the index procedure, follow-up data were collected via face-to-face interviews at the outpatient clinic, review of the medical records, and/or telephone contact with the patients. The primary clinical endpoints were the target extremity amputation and revascularization at one year. The secondary clinical endpoint was as follows: major adverse cardiac and cerebrovascular events (MACCEs), defined as the composite of total death, recurrent myocardial infarction, strokes, percutaneous coronary intervention, and coronary artery bypass graft.

## Statistical analysis

Statistical analysis was performed using SPSS software (Version 20.0, SPSS Inc., Chicago, IL, USA). All data are expressed as the mean ± standard deviation. Unpaired Student's t-test and Mann–Whitney rank test were used to compare continuous variables. The chi-square test and

Fisher's exact test were used to compare categorical variables. To adjust for potential confounders, propensity score matching (PSM) analysis was performed using the logistic regression model. We tested all available variables that could be of potential relevance: age, male sex, cardiovascular risk factors (hypertension, diabetes, dyslipidemia, current smokers, and current alcoholics. The propensity score was estimated using the C-statistic for the logistic regression model; the C-statistic value was 0.710. Matching was performed using a 1:1 matching protocol without replacement (nearest neighbor matching algorithm), with a caliper width equal to 0.02 of the standard deviation of the logit of the propensity score. A value of $p<0.05$ was considered as statistically significant.

## Results

In the present study, a total of 357 patients who underwent PTA with stent implantation or POBA alone from September 2010 to December 2016 were enrolled (stent group: n = 102, POBA group: n = 255). After PSM analysis, two propensity-matched groups (93 pairs, total = 186) were generated (Table 1).

### Baseline clinical and laboratory characteristics

Baseline clinical and laboratory characteristics are shown in Tables 1 and 2. The baseline clinical and laboratory characteristics were similar between the two groups; the hemoglobin level was lower in the POBA group than in the stent group. However, after PSM, the difference was not significant (Table 1).

### Lesion, limbs, angiographic, and procedural characteristics

Among the 357 patients, a total of 400 limbs (stent group: n = 111 limbs, POBA group: n = 289 limbs) were treated. After PSM analysis, two propensity-matched groups (101 pairs, total = 202) were generated (Table 2). The characteristics of the limbs are shown in Table 2. The baseline characteristics of the limbs were similar between the two groups, except that combined iliac artery lesions were more frequent in the stent group. Nevertheless, after PSM, the difference was not significant (Table 2).

In the present study, a total of 654 lesions (stent lesion: n = 127, POBA lesion n = 527 lesion) were treated. After PSM analysis, two propensity-matched groups (stent lesion: n = 116, POBA lesion n = 329 lesion) were generated (Table 3). Ostial lesions were more frequent in the stent group, but the difference was not significant after PSM. The lesion length was longer in the POBA lesion group than in the stented lesion group. The reference vessel diameter, minimum lumen diameter, and acute gain were larger in the stent group. Postprocedural diameter stenosis was smaller in the stent group. After PSM analysis, the results were consistent (Table 3). Procedural complications were similar in both groups, as demonstrated in Table 4.

### Six to 9-month follow-up imaging and 1-year clinical outcomes

Follow-up angiography was performed in a total of 146 limbs (the stent group: n = 49 limbs, the POBA group: n = 97 limbs). After PSM analysis, data for a total of 78 limbs were generated (Table 5). The rates of binary restenosis, primary patency, and secondary patency were similar between the two groups. The 1-year clinical outcomes were not different between the two groups, except target extremity revascularization (TER), which was more frequent in the stent group before matching. However, after PSM analysis, the differences were not statistically significant (Table 6).

**Table 1. Baseline clinical and laboratory characteristics.**

| Variables, n (%) | All Patients | | | Matched Patients | | |
|---|---|---|---|---|---|---|
| | Stent (n = 102) | POBA (n = 255) | *p* value | Stent (n = 93) | POBA (n = 93) | p value |
| Sex, male | 81 (79.4) | 199 (78.0) | 0.776 | 75 (80.6) | 74 (79.6) | 0.854 |
| Age, year | 67.8 ± 10.6 | 69.7 ± 12.6 | 0.153 | 68.7 ± 10.8 | 69.7 ± 13.5 | 0.424 |
| Body mass index, kg/m$^2$ | 23 ± 3 | 23.7 ± 4 | 0.404 | 23.0 ± 3.1 | 23.0 ± 4.1 | 0.536 |
| **Blood pressure, mmHg** | | | | | | |
| Systolic- | 147 ± 24 | 145 ± 23 | 0.330 | 148 ± 23 | 144 ± 24 | 0.509 |
| Diastolic- | 73 ± 40 | 68 ± 14 | 0.129 | 72 ± 14 | 69 ± 16 | 0.547 |
| Heart rate, bpm | 78 ± 13 | 81 ± 14 | 0.130 | 79 ± 13 | 82 ± 14 | 0.607 |
| LV ejection fraction, % | 60.3 ± 9.0 | 61.9 ± 8.3 | 0.090 | 59.8 ± 9.2 | 63.2 ± 4.7 | 0.057 |
| **Initial diagnosis** | | | | | | |
| Diabetic foot ulcer | 74 (72.5) | 191 (74.9) | 0.646 | 66 (71.0) | 63 (67.7) | 0.633 |
| Wound | 79 (77.5) | 210 (82.4) | 0.287 | 71 (76.3) | 72 (77.4) | 0.862 |
| Gangrene | 40 (39.2) | 106 (41.6) | 0.683 | 38 (40.9) | 37 (39.8) | 0.881 |
| Claudication | 14 (13.7) | 37 (14.5) | 0.848 | 13 (14.0) | 15 (16.1) | 0.682 |
| Resting pain | 6 (5.9) | 12 (4.7) | 0.646 | 6 (6.5) | | > 0.99 |
| Berger's disease | 3 (2.9) | 12 (4.7) | 0.569 | 3 (3.2) | 3 (3.2) | > 0.99 |
| Other | 2 (2.0) | 1 (0.4) | 0.198 | 2 (2.2) | 1 (1.1) | > 0.99 |
| **Patients history** | | | | | | |
| Hypertension | 69 (67.6) | 180 (70.6) | 0.585 | 62 (66.7) | 60 (64.5) | 0.758 |
| Diabetes mellitus | 85 (83.3) | 216 (84.7) | 0.747 | 77 (82.8) | 74 (79.6) | 0.574 |
| Insulin | 31 (30.4) | 106 (41.6) | 0.050 | 28 (30.1) | 30 (32.3) | 0.752 |
| Oral medication | 35 (34.3) | 80 (31.4) | 0.591 | 31 (33.3) | 31 (33.3) | > 0.99 |
| Untreated or diet | 3 (2.9) | 9 (3.5) | > 0.99 | 3 (3.2) | 4 (4.3) | > 0.99 |
| Dyslipidemia | 10 (9.8) | 19 (7.5) | 0.462 | 9 (9.7) | 7 (7.5) | 0.601 |
| Strokes | 16 (15.7) | 50 (19.6) | 0.389 | 15 (16.1) | 7 (7.5) | 0.069 |
| Hemorrhagic | 3 (2.9) | 4 (1.6) | 0.412 | 3 (3.2) | 1 (1.1) | 0.621 |
| Ischemic | 13 (12.7) | 46 (18) | 0.224 | 12 (12.9) | 6 (6.5) | 0.137 |
| Chronic renal insufficiency | 34 (33.3) | 107 (42) | 0.132 | 31 (33.3) | 28 (30.1) | 0.636 |
| Dialysis | 18 (17.6) | 66 (25.9) | 0.097 | 17 (18.3) | 18 (19.4) | 0.851 |
| Congestive heart failure | 6 (5.9) | 17 (6.7) | 0.785 | 5 (5.4) | 4 (4.3) | > 0.99 |
| Atrial fibrillation | 10 (9.8) | 27 (10.6) | 0.826 | 9 (9.7) | 9 (9.7) | > 0.99 |
| History of smoking | 54 (52.9) | 132 (51.8) | 0.841 | 50 (53.8) | 47 (50.5) | 0.660 |
| Current cigarette consumer | 29 (28.4) | 79 (31.0) | 0.636 | 26 (28.0) | 30 (32.3) | 0.523 |
| Past alcohol drinking | 35 (34.3) | 92 (36.1) | 0.753 | 32 (34.4) | 36 (38.7) | 0.543 |
| Current consumer of alcohol | 20 (19.6) | 47 (18.4) | 0.797 | 18 (19.4) | 21 (22.6) | 0.589 |
| Significant stenosis (> 70%) | 49 (48.0) | 140 (54.9) | 0.241 | 44 (47.3) | 39 (41.9) | 0.461 |
| Treated CAD | 32 (31.4) | 104 (40.8) | 0.098 | 30 (32.3) | 21 (22.6) | 0.139 |
| CABG | 4 (3.9) | 11 (4.3) | > 0.99 | 4 (4.3) | 1 (1.1) | 0.368 |
| PCI | 31 (30.4) | 100 (39.2) | 0.118 | 29 (31.2) | 21 (22.6) | 0.186 |
| **Laboratory findings** | | | | | | |
| Hemoglobin, mg/dL | 11.4 ± 2.0 | 10.7 ± 1.7 | 0.029 | 11.6 ± 2.1 | 10.8 ± 1.6 | 0.165 |
| Fasting glucose, mg/dL | 146.7 ± 72.8 | 167.9 ± 101.8 | 0.309 | 134.7 ± 62.8 | 163.3 ± 109.2 | 0.327 |
| Hemoglobin A1c, % | 7.25 ± 1.51 | 7.5 ± 1.84 | 0.504 | 7.06 ± 1.49 | 7.2 ± 1.78 | 0.738 |
| high-sensitivity CRP | 25.5 ± 44.1 | 35.9 ± 41.3 | 0.542 | 19.7 ± 25.0 | 31.0 ± 39.5 | 0.950 |
| Creatinine, mg/dL | 2.44 ± 2.8 | 2.46 ± 2.58 | 0.282 | 2.06 ± 2.5 | 2.18 ± 2.56 | 0.248 |
| **Post-procedural medications** | | | | | | |
| Aspirin | 100 (98.0) | 249 (97.6) | > 0.99 | 91 (97.8) | 91 (97.8) | > 0.99 |

(*Continued*)

**Table 1.** (Continued)

| Variables, n (%) | All Patients | | | Matched Patients | | |
|---|---|---|---|---|---|---|
| | Stent (n = 102) | POBA (n = 255) | p value | Stent (n = 93) | POBA (n = 93) | p value |
| Clopidogrel | 95 (93.1) | 224 (87.8) | 0.143 | 86 (92.5) | 84 (90.3) | 0.601 |
| Cilostazol | 39 (38.2) | 80 (31.4) | 0.214 | 37 (39.8) | 32 (34.4) | 0.448 |
| Ticlopidine | 0 (0.0) | 2 (0.8) | > 0.99 | | | |
| Sarpogrelate | 10 (9.8) | 9 (3.5) | 0.017 | 8 (8.6) | 3 (3.2) | 0.120 |
| ARBs | 46 (45.1) | 93 (36.5) | 0.131 | 41 (44.1) | 34 (36.6) | 0.295 |
| ACEI | 12 (11.8) | 20 (7.8) | 0.241 | 11 (11.8) | 8 (8.6) | 0.468 |
| CCB | 41 (40.2) | 106 (41.6) | 0.812 | 39 (41.9) | 34 (36.6) | 0.453 |
| β-blocker | 23 (22.5) | 68 (26.7) | 0.420 | 20 (21.5) | 20 (21.5) | > 0.99 |
| Diuretics | 24 (23.5) | 50 (19.6) | 0.409 | 21 (22.6) | 15 (16.1) | 0.265 |
| Statin | 88 (86.3) | 223 (87.5) | 0.764 | 81 (87.1) | 79 (84.9) | 0.672 |

LV: left ventricle; CAD: coronary artery disease; CABG: coronary artery bypass graft; PCI: percutaneous coronary intervention; CRP: c–reactive protein; ARB: angiotensin receptor blocker; ACE: angiotensin converting enzyme; CCB: calcium channel blockers.

## Discussion

The main findings of this study were as follows: in CLI patients undergoing BTK PTA, 1) cardiovascular clinical outcomes, including cardiac death and myocardial infarction, were similar between the stenting and POBA groups; 2) target lesion-related 1-year clinical outcomes and angiographic outcomes were similar between the stenting and POBA groups; except for a trend toward a higher incidence of TER rate in the stenting group; and 3) a relatively higher limb salvage rate was observed, despite the lower primary patency and higher revascularization rates with optimal medical therapy, rehabilitation, and multidisciplinary approaches for CLI patients. Compared with previous BTK studies, we recruited a relatively larger number of patients in this study.

Mortality and cardiovascular adverse events are the most important parameters to justify a study in cardiovascular areas, including the treatment of PAD. Recently, some treatment modalities, such as drug-coated balloons (DCBs), have shown worse long-term mortality in the treatment of PAD, despite improvements in the patency rate and amputation-free survival rate [16, 17]. In our study, very low rates of cardiac death were observed in both the stent and the POBA groups, and no differences in mortality or cardiovascular adverse outcomes, such as major adverse cardiac and cerebrovascular events, were noted between the two groups. In addition, procedural complication rates were low and identical in both groups. We thought that the tendency of high TER was observed numerically before matching. However, after PSM analysis, the differences between the two groups were not statistically significant. Considering that one previous study reported a high mortality rate of approximately 25%, our study showed relatively low mortality rates in both groups. Based on our study results, both strategies can be performed safely in the treatment of BTK lesions.

In interventions in the BTK area, various therapeutic endpoints are applied, such as wound healing (limb salvage, pain relief, quality of life improvement, and patency rate [including the quantification of restenosis]) [18–20]. Through this study, we also tried to focus on the rate of revascularization due to restenosis and the limb salvage rate, including the amputation rate.

In our study, the tendency of a high TER rate was observed numerically before matching. However, after PSM analysis, the differences between the two groups were not statistically significant. The most important indicators among the various purposes of BTK intervention are the limb salvage rate and the revascularization rate [21–23]. The results of this study suggest

**Table 2. Baseline angiographic and clinical characteristics of the patients' limbs.**

| Variables, n (%) | All Patients | | | Matched Patients | | |
|---|---|---|---|---|---|---|
| | Stent (111 Limbs) | POBA (289 Limbs) | *p* value | Stent (101 Limbs) | POBA (101 Limbs) | *p* value |
| **Limb site** | | | | | | |
| Right | 56 (50.5) | 150 (51.9) | 0.795 | 47 (46.5) | 39 (38.6) | 0.255 |
| Left | 55 (49.5) | 139 (48.1) | 0.795 | 54 (53.5) | 62 (61.4) | 0.255 |
| **Ankle brachial pressure index** | 0.80 ± 0.29 | 0.82 ± 0.33 | 0.841 | 0.81 ± 0.3 | 0.83 ± 0.35 | 0.777 |
| **Rutherford grade, Limb** | | | | | | |
| Grade 0 (Category 0) | 3 (2.7) | 10 (3.5) | 0.451 | 3 (3.0) | 2 (2.0) | 0.811 |
| Grade 1 | 24 (21.6) | 46 (15.9) | | 23 (22.8) | 19 (18.8) | |
| Category 1 | 1 (0.9) | 7 (2.4) | | 1 (1.0) | 2 (2.0) | |
| Category 2 | 6 (5.4) | 8 (2.8) | | 5 (5.0) | 4 (4.0) | |
| Category 3 | 17 (15.3) | 31 (10.7) | | 17 (16.8) | 13 (12.9) | |
| Grade 2 | 54 (48.6) | 162 (56.1) | | 47 (46.5) | 53 (52.5) | |
| Category 4 | 13 (11.7) | 26 (9.0) | | 13 (12.9) | 11 (10.9) | |
| Category 5 | 41 (36.9) | 136 (47.1) | | 34 (33.7) | 42 (41.6) | |
| Grade 3 (Category 6) | 30 (27.0) | 71 (24.6) | | 28 (27.7) | 27 (26.7) | |
| **Location, Limb** | | | | | | |
| distal Aorta | 1 (0.9) | 0 (0.0) | 0.277 | 1 (1.0) | 0 (0.0) | > 0.99 |
| **Illiac** | 15 (13.5) | 17 (5.9) | 0.012 | 13 (12.9) | 13 (12.9) | > 0.99 |
| Femoral | 47 (42.3) | 113 (39.1) | 0.553 | 43 (42.6) | 40 (39.6) | 0.668 |
| Popliteal | 13 (11.7) | 27 (9.3) | 0.479 | 11 (10.9) | 10 (9.9) | 0.818 |
| **Tibio-peroneal trunk** | 6 (5.4) | 5 (1.7) | 0.079 | 5 (5.0) | 3 (3.0) | 0.721 |
| Anterior tibial artery | 94 (84.7) | 234 (81) | 0.386 | 84 (83.2) | 79 (78.2) | 0.373 |
| Posterior tibial artery | 69 (62.2) | 187 (64.7) | 0.635 | 59 (58.4) | 59 (58.4) | > 0.99 |
| Peroneal artery | 43 (38.7) | 94 (32.5) | 0.241 | 36 (35.6) | 30 (29.7) | 0.368 |
| **Pre-amputations, number of patients** | | | | | | |
| before admission for PTA | 8 (7.2) | 28 (9.7) | 0.437 | 7 (6.9) | 7 (6.9) | > 0.99 |
| Major, Above the ankle | 1 (0.9) | 0 (0.0) | 0.277 | 1 (1.0) | 0 (0.0) | > 0.99 |
| minor, Below the ankle | 9 (8.1) | 28 (9.7) | 0.625 | 8 (7.9) | 7 (6.9) | 0.788 |
| **On admission before PTA** | 43 (38.7) | 140 (48.4) | 0.081 | 40 (39.6) | 49 (48.5) | 0.202 |
| Target extremity surgery | | | | | | |
| Major, Above the ankle | 20 (18.0) | 61 (21.1) | 0.491 | 19 (18.8) | 19 (18.8) | > 0.99 |
| minor, Below the ankle | 0 (0.0) | 1 (0.3) | > 0.99 | | | |
| **Ostectomy (not included joint)** | 20 (18.0) | 59 (20.4) | 0.590 | 19 (18.8) | 19 (18.8) | > 0.99 |

PTA: percutaneous transluminal angioplasty.

that BMS implantation remains a necessary treatment option relative to POBA, although the TER rate tended to be higher in the stenting group than in the POBA group, especially in the bail-out situation.

In previous studies, high restenosis, a low primary patency rate, and a high TER rate were the main problems, despite the relatively high limb salvage rate in the interventional treatment of BTK lesions [4, 24]. Our study showed that target extremity amputation rates were low in both groups (14.4% in the stent group and 19.0% in the POBA group). In particular, the TER rate was significantly higher in the stenting group than in the POBA group before matching. In the previous studies, the proportion of TLRs was widely formed, but most were reported to be over 15% [18, 24, 25]. However, the absolute percentages of TLRs were lower than those reported in previous studies [18, 24, 25]. These results are thought to be due to the fact that

**Table 3. Baseline angiographic and clinical characteristics of the patients' lesions during procedures.**

| | All Patients | | | Matched Patients | | |
|---|---|---|---|---|---|---|
| | Stent (127 Lesion) | POBA (527 Lesion) | *p* value | Stent (116 Lesion) | POBA (329 Lesion) | *p* value |
| **Lesion site** | | | | | | |
| Tibio-peroneal trunk | 4 (3.1) | 5 (0.9) | 0.106 | 4 (3.4) | 3 (1.8) | 0.368 |
| Anterior tibial artery | 69 (54.3) | 237 (45.0) | 0.052 | 62 (53.4) | 79 (46.2) | 0.077 |
| Posterior tibial artery | 40 (31.5) | 191 (36.2) | 0.564 | 38 (32.8) | 59 (34.5) | 0.952 |
| Peroneal artery | 14 (11.0) | 94 (17.8) | 0.005 | 12 (10.3) | 30 (17.5) | 0.003 |
| **Lesion location** | | | | | | |
| Ostial | 35 (27.6) | 75 (14.2) | 0.001 | 30 (25.9) | 38 (22.2) | 0.776 |
| Proximal | 67 (52.8) | 320 (60.7) | 0.257 | 62 (53.4) | 85 (49.7) | 0.356 |
| Mid | 17 (13.4) | 86 (16.3) | 0.213 | 16 (13.8) | 32 (18.7) | 0.146 |
| Distal | 8 (6.3) | 46 (8.7) | 0.640 | 8 (6.9) | 16 (9.4) | 0.731 |
| **Lesion type** | | | | | | |
| Concentric | 31 (24.4) | 101 (19.2) | 0.294 | 29 (25) | 33 (19.3) | 0.417 |
| Eccentric | 10 (7.9) | 48 (9.1) | 0.574 | 8 (6.9) | 10 (5.8) | 0.870 |
| **Total occlusion** | 83 (65.4) | 370 (70.2) | 0.551 | 77 (66.4) | 125 (73.1) | 0.422 |
| **Chronic total occlusion** | 69 (54.3) | 308 (58.4) | 0.682 | 64 (55.2) | 109 (63.7) | 0.277 |
| **Quantitative angiography** | | | | | | |
| **Lesion length, mm** | 51 ± 27 | 163 ± 82 | < 0.01 | 48 ± 19 | 165 ± 87 | < 0.01 |
| **Reference vessel diameter, mm** | | | | | | |
| pre | 3.4 ± 0.71 | 2.56 ± 0.49 | < 0.01 | 3.39 ± 0.67 | 2.58 ± 0.43 | < 0.01 |
| post | 3.5 ± 0.64 | 2.64 ± 0.51 | < 0.01 | 3.51 ± 0.58 | 2.69 ± 0.49 | < 0.01 |
| **Minimum lumen diameter, mm** | | | | | | |
| pre | 0.18 ± 0.33 | 0.18 ± 0.30 | 0.512 | 0.17 ± 0.32 | 0.12 ± 0.25 | 0.421 |
| post | 2.9 ± 0.59 | 1.99 ± 0.69 | < 0.01 | 2.96 ± 0.51 | 1.95 ± 0.75 | < 0.01 |
| **Diameter stenosis, %** | | | | | | |
| pre | 94 ± 9 | 92 ± 12 | 0.262 | 94 ± 9 | 94 ± 10 | 0.933 |
| post | 13 ± 11 | 22 ± 21 | < 0.01 | 12 ± 6 | 24 ± 26 | < 0.01 |
| **Acute gain, mm** | 2.7 ± 0.66 | 1.80 ± 0.71 | < 0.01 | 2.77 ± 0.58 | 1.79 ± 0.77 | < 0.01 |
| **Sub-intimal approach** | 38 (29.9) | 167 (31.7) | 0.333 | 35 (30.2) | 60 (35.1) | 0.442 |
| **Stents type** | | | | | | |
| Expert | 72 (56.7) | - | - | 67 (57.8) | - | - |
| Maris deep | 44 (34.6) | - | - | 42 (36.2) | - | - |
| Chromis deep | 11 (8.7) | - | - | 7 (6.0) | - | - |
| **Complication** | | | | | | |
| Dissection | 35 (27.6) | 86 (16.3) | < 0.01 | 32 (27.6) | 27 (15.8) | 0.001 |
| Acute thrombosis | 5 (3.9) | 6 (1.1) | 0.089 | 4 (3.4) | 2 (1.2) | 0.419 |
| Abrupt closure | 2 (1.6) | 2 (0.4) | 0.243 | 2 (1.7) | 1 (0.6) | 0.488 |
| No-reflow | 0 (0.0) | 3 (0.6) | 0.615 | 0 (0.0) | 1 (0.6) | 0.629 |
| Perforation | 1 (0.8) | 30 (5.7) | 0.011 | 1 (0.9) | 14 (8.2) | 0.009 |
| Rupture | 1 (0.8) | 21 (4) | 0.086 | 1 (0.9) | 8 (4.7) | 0.077 |

reintervention is not performed unless the wound recurs as the aim of the intervention of the BTK lesion is shifted to the clinical goal of limb salvage more than anatomical restenosis and the development of techniques and advancement of the operator's experience.

In the present study, the 1-year clinical outcomes between the two groups were not significantly different in terms of amputation rate and cardiovascular clinical outcomes, including mortality. In addition, our data showed a higher trend in the incidence of TER in the stenting

**Table 4. Access site and in–hospital complications.**

| Variables, n (%) | All Patients | | | Matched Patients | | |
|---|---|---|---|---|---|---|
| | Stent (n = 102) | POBA (n = 255) | *p* value | Stent (n = 93) | POBA (n = 93) | *p*-value |
| **Complications at access site** | | | | | | |
| **Arteriovenous fistula** | 0 (0.0) | 1 (0.4) | > 0.99 | - | - | - |
| **Pseudoaneurysm** | 0 (0.0) | 3 (1.2) | 0.561 | - | - | - |
| **Hematoma** | 5 (4.9) | 17 (6.7) | 0.531 | 5 (5.4) | 4 (4.3) | > 0.99 |
| Minor, < 4 cm | 3 (2.9) | 9 (3.5) | > 0.99 | 3 (3.2) | 2 (2.2) | > 0.99 |
| Major, > 4 cm | 2 (2.0) | 8 (3.1) | 0.731 | 2 (2.2) | 2 (2.2) | > 0.99 |
| **Bleeding complications** | 18 (17.6) | 44 (17.3) | 0.930 | 16 (17.2) | 12 (12.9) | 0.412 |
| **Major bleeding** | 3 (2.9) | 3 (1.2) | 0.359 | 3 (3.2) | 0 (0.0) | 0.246 |
| Gastrointestinal bleeding | 0 (0.0) | 3 (1.2) | 0.561 | - | - | - |
| Retroperitoneal bleeding | 3 (2.9) | 0 (0.0) | 0.023 | 3 (3.2) | 0 (0.0) | 0.246 |
| **Transfusion** | 29 (28.4) | 77 (30.2) | 0.742 | 27 (29.0) | 26 (28.0) | 0.871 |
| Transfusion, pint | 2.0 ± 5.7 | 3.2 ± 8.0 | 0.114 | 1.7 ± 4.8 | 3.4 ± 8.7 | 0.122 |
| **In-hospital complications** | | | | | | |
| **Acute limb ischemia** | 1 (1.0) | 2 (0.8) | > 0.99 | 1 (1.1) | 1 (1.1) | > 0.99 |
| **Acute renal failure** | 0 (0.0) | 5 (2.0) | 0.327 | 0 (0.0) | 1 (1.1) | > 0.99 |
| **Congestive heart failure** | 0 (0.0) | 2 (0.8) | > 0.99 | 0 (0.0) | 1 (1.1) | > 0.99 |
| **Strokes** | 0 (0.0) | 2 (0.8) | > 0.99 | - | - | - |
| Ischemic | 0 (0.0) | 2 (0.8) | > 0.99 | - | - | - |

group than in the POBA group, but the results were not statistically significant after propensity matching. In addition, a numerically higher limb salvage rate was shown in stent group compared to POBA group, so stent implantation may remain a treatment option in special circumstances, such as flow limiting dissection and persistent recoil despite of repeated balloon dilatation. This study has some limitations. First, the present study was designed as a nonrandomized, single-center study, but the data were collected prospectively. Thus, the results are subject to potential bias and must be interpreted carefully. Second, follow-up imaging modalities, such as CT angiography and conventional angiography, were not routinely performed in all patients. In particular, operators decided that follow-up angiography mainly depended on the recurrence of wounds or the physician's discretion, regardless of ischemic symptoms such as claudication. If restenosis occurred, but the wound did not recur, follow-up imaging tests may not have been performed, as they do not reflect the actual restenosis rate. Third, the relatively small sample size and short follow-up period are not a strong proof of the actual clinical

**Table 5. Six–to 9– month angiographic outcomes.**

| Variables, n (%) | Stent (49 Limbs) | POBA (97 Limbs) | *P* value | Stent (45 Limbs) | POBA (33 Limbs) | *P* value |
|---|---|---|---|---|---|---|
| **Angiography** | 49 (100.0) | 97 (100.0) | - | 45 (100.0) | 33 (100.0) | - |
| **CT angiography** | 17 (34.7) | 39 (40.2) | 0.518 | 15 (33.3) | 15 (45.5) | 0.277 |
| **Doppler** | 2 (4.1) | 6 (6.2) | 0.718 | 0 (0.0) | 2 (6.1) | 0.176 |
| **Binary restenosis** | 34 (69.4) | 67 (69.1) | 0.969 | 31 (67.7) | 23 (69.7) | 0.939 |
| **Total occlusion** | 22 (44.9) | 55 (56.7) | 0.177 | 21 (46.7) | 22 (66.7) | 0.079 |
| **Primary patency** | 15 (30.6) | 30 (30.9) | 0.969 | 14 (31.1) | 10 (30.3) | 0.939 |
| Assisted primary patency | 20 (40.8) | 31 (32) | 0.289 | 19 (42.2) | 10 (30.3) | 0.282 |
| **Secondary patency** | 39 (79.6) | 68 (70.1) | 0.221 | 36 (80.0) | 22 (66.7) | 0.183 |

CT: computed tomography.

**Table 6. One–year clinical outcomes.**

| Variables, n (%) | Stent (n = 102) | POBA (n = 255) | P value | Stent (n = 93) | POBA (n = 93) | P value |
|---|---|---|---|---|---|---|
| **One-year clinical outcomes** | | | | | | |
| **Total death** | 8 (7.8) | 19 (7.5) | 0.899 | 8 (8.6) | 4 (4.3) | 0.233 |
| Cardiac death | 1 (1.0) | 3 (1.2) | > 0.99 | 1 (1.1) | 0 (0.0) | > 0.99 |
| **MI** | 1 (1.0) | 2 (0.8) | > 0.99 | 1 (1.1) | 0 (0.0) | > 0.99 |
| STEMI | 0 (0.0) | 1 (0.4) | > 0.99 | - | - | - |
| **PCI** | 4 (3.9) | 6 (2.4) | 0.480 | 4 (4.3) | 1 (1.1) | 0.368 |
| Target vessel | 3 (2.9) | 4 (1.6) | 0.412 | 3 (3.2) | 0 (0.0) | 0.246 |
| **Strokes** | 2 (2.0) | 8 (3.1) | 0.731 | 2 (2.2) | 1 (1.1) | > 0.99 |
| **Major adverse cardiac and cerebrovascular events** | 12 (11.8) | 28 (11.0) | 0.832 | 12 (12.9) | 6 (6.5) | 0.137 |
| Variables, n (%) | Stent (111 Limbs) | POBA (289 Limbs) | P value | Stent (101 Limbs) | POBA (101 Limbs) | P value |
| **One-year clinical outcomes** | | | | | | |
| **Peripheral revascularization** | | | | | | |
| Target lesion - | 15 (13.5) | 30 (10.4) | 0.375 | 15 (14.9) | 11 (10.9) | 0.401 |
| Target extremity - | 21 (18.9) | 32 (11.1) | 0.038 | 21 (20.8) | 11 (10.9) | 0.054 |
| **Target extremity amputations** | 16 (14.4) | 55 (19.0) | 0.279 | 16 (15.8) | 22 (21.8) | 0.280 |
| Above the knee | 1 (0.9) | 1 (0.3) | 0.478 | 1 (1.0) | 1 (1.0) | > 0.99 |
| Above the ankle | 4 (3.6) | 17 (5.9) | 0.360 | 4 (4.0) | 8 (7.9) | 0.234 |
| Below the ankle | 16 (14.4) | 54 (18.7) | 0.314 | 16 (15.8) | 21 (20.8) | 0.363 |

PCI: percutaneous coronary intervention; MI: myocardial infarction; STEMI: ST–segment elevation myocardial infarction; PCI: percutaneous coronary intervention

efficacy and safety between the two groups. Finally, the stenting group had heterogeneous treatment strategies (primary versus provisional) and heterogeneous types of stents. Thus, to increase the credibility of the research, it is necessary to unify the methodology of the treatment and the type of stent. Despite its limitations, this study investigated patients with CLI who were prospectively enrolled. However, long-term studies with larger numbers of patients with unified treatment methodologies are required to reach a final conclusion.

## Conclusions

Although there was a trend toward a higher incidence of TER risk in the stenting group, stent implantation seemed to have acceptable 1-year safety and efficacy compared with POBA alone in CLI patients undergoing BTK PTA. These results indicate that stent implantation could remain the preferred strategy in special circumstances, such as flow-limiting dissection and recoil despite frequent prolonged balloon dilatation. Long-term randomized studies with larger study populations and unified methodologies are necessary to elucidate the final conclusion.

## Supporting information

**S1 Raw data.**
(XLSX)

## Author Contributions

**Conceptualization:** Jihun Ahn, Byoung Geol Choi, Jin Oh Na, Cheol Ung Choi, Eung Ju Kim, Chang Gyu Park, Hong Seog Seo, Woong-gil Choi, Seung-Woon Rha.

**Data curation:** Seung-Woon Rha.

**Formal analysis:** Jihun Ahn, Byoung Geol Choi, Se Yeon Choi, Jae Kyeong Byun, Jinah Cha, Seung-Woon Rha.

**Funding acquisition:** Seung-Woon Rha.

**Investigation:** Jihun Ahn, Jin Oh Na, Cheol Ung Choi, Eung Ju Kim, Chang Gyu Park, Hong Seog Seo, Woong-gil Choi, Seung-Woon Rha.

**Methodology:** Byoung Geol Choi, Cheolho Lee, Seung-Woon Rha.

**Project administration:** Seung-Woon Rha.

**Resources:** Jihun Ahn, HyeYon Yu, Seung-Woon Rha.

**Software:** Seung-Woon Rha.

**Supervision:** Jihun Ahn, HyeYon Yu, Seung-Woon Rha.

**Validation:** Jihun Ahn, Seung-Woon Rha.

**Visualization:** HyeYon Yu, Seung-Woon Rha.

**Writing – original draft:** Jihun Ahn, JinSu Byeon, Byoung Geol Choi, Seung-Woon Rha.

**Writing – review & editing:** Jihun Ahn, JinSu Byeon, Byoung Geol Choi, Seung-Woon Rha.

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
