## [Decision Letter · Decision Letter 0]

14 Apr 2021

PONE-D-21-00154

Stenting versus Balloon Angioplasty Alone in Patients with Below-the-Knee Disease: A Propensity Score-Matched Analysis

PLOS ONE

Dear Dr. Rha,

Thank you for submitting your manuscript to PLOS ONE. After careful consideration, we feel that it has merit but does not fully meet PLOS ONE’s publication criteria as it currently stands. Therefore, we invite you to submit a revised version of the manuscript that addresses the points raised during the review process.

Please submit your revised manuscript in 60 days. If you will need more time than this to complete your revisions, please reply to this message or contact the journal office at plosone@plos.org. Please include the following items when submitting your revised manuscript:

We look forward to receiving your revised manuscript.

Kind regards,

Timir Paul

Academic Editor

PLOS ONE

Journal Requirements:

No

2a)           Please clarify the sources of funding (financial or material support) for your study. List the grants or organizations that supported your study, including funding received from your institution.

2b)           State what role the funders took in the study. If the funders had no role in your study, please state: “The funders had no role in study design, data collection and analysis, decision to publish, or preparation of the manuscript.”

2c)           If any authors received a salary from any of your funders, please state which authors and which funders.

2d)           If you did not receive any funding for this study, please state: “The authors received no specific funding for this work.”

NO

4, Please amend your list of authors on the manuscript to ensure that each author is linked to an affiliation. Authors’ affiliations should reflect the institution where the work was done (if authors moved subsequently, you can also list the new affiliation stating “current affiliation:….” as necessary).

Reviewers' comments:

Reviewer's Responses to Questions

**Comments to the Author**

1. Is the manuscript technically sound, and do the data support the conclusions?

Reviewer #1: Yes

Reviewer #2: Yes

2. Has the statistical analysis been performed appropriately and rigorously? 

Reviewer #1: Yes

Reviewer #2: Yes

3. Have the authors made all data underlying the findings in their manuscript fully available?

Reviewer #1: Yes

Reviewer #2: Yes

4. Is the manuscript presented in an intelligible fashion and written in standard English?

Reviewer #1: Yes

Reviewer #2: Yes

5. Review Comments to the Author

Reviewer #1: The authors here did the propensity matched analysis of stenting versus balloon angioplasty in the critical limb ischemia (CLI) patients with below the knee disease. The optimal treatment of below the knee lesions either with balloon angioplasty or stenting remains controversial. The data is sparse with majority of studies with small sample size.

The rationale and careful nature of the data review and statistical methods appear reasonably sound. The study limitations have been pointed out correctly. The authors showed no difference in cardiovascular clinical outcomes and target lesion related 1 year clinical outcomes in balloon angioplasty versus stenting group after propensity matching.

- Please correct second last line on page 4, “believe” has been used twice.

Reviewer #2: This study aimed to evaluate the safety and efficacy of BMS compared to PTA in BTK lesions. It is a retrospective study that included 350+ patients that underwent either PTA or BMS for BTK. Propensity matching was performed to achieve matched groups. Both angiographic and clinical end points were analyzed. The authors conclude bare metal stents were similar in outcomes compared to PTA alone in these patients.

My comments are as follows:

1. The Cochrane review on this topic (Cochrane Database Syst Rev. 2018 Dec 8;12(12):CD009195) should be included either in the introduction section or discussion. This also concluded there is no significant added benefit of stenting compared to PTA alone for BTK.

2. The authors report the stent group was divided into elective or bailout stenting, although I do not see this difference reported in the results section. Were the results similar?

3. In the results section, page 11, it is reported that after PSM analysis, three matched groups of lesions were obtained. Yet in table 3 only 2 groups are reported. This needs clarification.

4. In table 3, both columns read "all patients" shouldn't one of them be PSM group?

5. Why were clinical outcomes studied as end point? How would stenting versus PTA alone for BTK affect MACCE?

6. The authors conclude that stenting is comparable to PTA alone for BTK in regards to angiographic end points (re-stenosis), yet it should be stressed that despite higher acute luminal gain and less percentage residual stenosis with stents, the repeat revascularization rates were similar or higher in stent group. This to me should be interpreted as PTA appears to be sufficient for these lesions, with stents to be used in very select group of patients. Not the other way around.

6. PLOS authors have the option to publish the peer review history of their article (what does this mean?). If published, this will include your full peer review and any attached files.

Reviewer #1: **Yes: **Sukhdeep Bhogal

Reviewer #2: **Yes: **Madhan Sundaram

---

## [Author Response · Author response to Decision Letter 0]

28 Apr 2021

RE: PONE-D-21-00154

Stenting versus Balloon Angioplasty Alone in Patients with Below-the-Knee Disease: A Propensity Score-Matched Analysis

PLOS ONE

We have tried our best to make revisions to improve the quality of the manuscript to editor comment. 

To make it easier for you to follow how changes were made, we have revised manuscript with red or blue color. 

Please consider the following attached files of the authors’ response and revisions according to reviewer’ comments. 

Thank you for your advice.

Sincerely 

From SW Rha

---

## [Editor Report · Decision Letter 1]

3 May 2021

Stenting versus Balloon Angioplasty Alone in Patients with Below-the-Knee Disease: A Propensity Score-Matched Analysis

PONE-D-21-00154R1

Dear Dr. Rha,

We’re pleased to inform you that your manuscript has been judged scientifically suitable for publication and will be formally accepted for publication once it meets all outstanding technical requirements.

Kind regards,

Timir Paul

Academic Editor

PLOS ONE
---

## [Editor Report · Acceptance letter]

2 Jun 2021

PONE-D-21-00154R1 

Stenting versus balloon angioplasty alone in patients with below-the-knee disease: A propensity score-matched analysis 

Dear Dr. Rha:

I'm pleased to inform you that your manuscript has been deemed suitable for publication in PLOS ONE. Congratulations! Your manuscript is now with our production department. 

Kind regards, 

on behalf of

Dr. Timir Paul 

Academic Editor

PLOS ONE